# Phylogenetic Diversity of Lhr Proteins and Biochemical Activities of the Thermococcales aLhr2 DNA/RNA Helicase

**DOI:** 10.3390/biom11070950

**Published:** 2021-06-26

**Authors:** Mirna Hajj, Petra Langendijk-Genevaux, Manon Batista, Yves Quentin, Sébastien Laurent, Régine Capeyrou, Ziad Abdel-Razzak, Didier Flament, Hala Chamieh, Gwennaele Fichant, Béatrice Clouet-d’Orval, Marie Bouvier

**Affiliations:** 1Laboratoire de Microbiologie et Génétique Moléculaires, UMR5100, Centre de Biologie Intégrative (CBI), Université de Toulouse, CNRS, Université Paul Sabatier, F-31062 Toulouse, France; mirna.hajj@univ-tlse3.fr (M.H.); petra.langendijk-genevaux@univ-tlse3.fr (P.L.-G.); manon.batista@univ-tlse3.fr (M.B.); yves.quentin@univ-tlse3.fr (Y.Q.); marie.bouvier@univ-tlse3.fr (M.B.); 2Laboratory of Applied Biotechnology (LBA3B), Azm Center for Research in Biotechnology and Its Application, EDST, Lebanese University, Tripoli 1300, Lebanon; ziad.abdelrazzak@ul.edu.lb (Z.A.-R.); hala.chamieh@ul.edu.lb (H.C.); 3Ifremer, Univ Brest, CNRS, UMR 6197 Laboratoire de Microbiologie des Environnements Extrêmes, F-29280 Plouzané, France; Sebastien.Laurent@ifremer.fr (S.L.); Didier.Flament@ifremer.fr (D.F.); 4Laboratoire de Biologie Moléculaire, Cellulaire et du Développement, UMR5077, Centre de Biologie Intégrative (CBI), Université de Toulouse, CNRS, Université Paul Sabatier, F-31062 Toulouse, France; regine.capeyrou@univ-tlse3.fr

**Keywords:** SF2 helicases, aLhr2 helicases, archaea, RNA metabolism, DNA repair, Thermococcales

## Abstract

Helicase proteins are known to use the energy of ATP to unwind nucleic acids and to remodel protein-nucleic acid complexes. They are involved in almost every aspect of DNA and RNA metabolisms and participate in numerous repair mechanisms that maintain cellular integrity. The archaeal Lhr-type proteins are SF2 helicases that are mostly uncharacterized. They have been proposed to be DNA helicases that act in DNA recombination and repair processes in Sulfolobales and Methanothermobacter. In Thermococcales, a protein annotated as an Lhr2 protein was found in the network of proteins involved in RNA metabolism. To investigate this, we performed in-depth phylogenomic analyses to report the classification and taxonomic distribution of Lhr-type proteins in Archaea, and to better understand their relationship with bacterial Lhr. Furthermore, with the goal of envisioning the role(s) of aLhr2 in Thermococcales cells, we deciphered the enzymatic activities of aLhr2 from *Thermococcus barophilus* (*Tbar*). We showed that *Tbar*-aLhr2 is a DNA/RNA helicase with a significant annealing activity that is involved in processes dependent on DNA and RNA transactions.

## 1. Introduction

Helicases are proteins that unwind nucleic acids and remodel protein-nucleic acid complexes in a wide spectrum of cellular tasks. DNA helicases are critical in maintaining cellular integrity by playing important roles in DNA replication, recombination and repair. RNA helicases are likewise fundamental by orchestrating transcription, RNA processing, ribosome biogenesis, translation and RNA turnover. Helicases are classified into 6 superfamilies (SF1-6) [1]. The SF1-6 share a common helicase core with a set of helicase signature motifs. The SF2 is the largest and most diverse group of helicases with more than ten families. SF2 members are non-hexameric helicases that share a conserved helicase core with nine characteristic motifs and that often contain N- and/or C-terminal accessory domains involved in the regulation of their activities [2,3]. The core provides the active site for ATP hydrolysis, binds nucleic acid and performs a basal unwinding activity. Although ATP-dependent unwinding of nucleic acid duplexes is their hallmark reaction, not all helicases catalyse unwinding in vitro, and disrupt duplexes in vivo [4,5]. Among SF2 helicases, the Lhr (Large helicase related) proteins are scarcely characterized. They are found in some Bacteria but are ubiquitous in Archaea [4,5]. To date, no homologs of Lhr proteins have been reported in Eukarya.

In Bacteria, Lhr proteins are mostly prevalent in Proteobacteria and Actinobacteria. Lhr proteins from *Pseudomonas putida* (*Pput*), *Escherichia coli* (*Ecol*) and *Mycobacterium smegmatis* (*Msme*) are among the few helicases that have been characterized [6,7,8,9]. The 1507 amino acid (aa) *Msme*-Lhr is the founding member of the Lhr helicase family [9]. The crystal structure of *Msme*-Lhr restricted to the first 856 aa was solved and uncovered a specific structural domain organization also referred to as the “Lhr-Core”: two RecA domains in tandem (RecA1 and RecA2), a winged-helix (WH) motif and a domain annotated as Domain 4 whose function is still unknown. Interestingly, the WH displays a similar fold to the one observed in Hjm and RecQ (also called Hel308) DNA helicases [9,10]. While *Pput*-Lhr is restricted to the “Lhr-Core”, *Msme*-Lhr and *Ecol*-Lhr have an additional C-terminal domain [6,7,8,9] (Figure 1).

The studies characterizing the biochemical activities and the functions of bacterial Lhr proteins have mainly revealed a role of Lhr helicases in DNA repair. Nonetheless, some of their properties suggest that Lhr may also participate in RNA processing. In vivo, the gene encoding *Msme*-Lhr was shown to be upregulated when cells were exposed to DNA damaging agents [11,12]. Regarding *Ecol*-Lhr, though its deletion does not increase cell-sensitivity to UV or H_2_O_2_ [8], Cooper et al. demonstrated a synthetic genetic interaction with RadA, a RecA-related protein involved in the processing of recombination intermediates [13]. In vitro, *Pput*-Lhr and *Msme*-Lhr helicases were shown to have DNA-dependent ATPase and ATP-dependent 3′-to-5′ translocase activities. While *Pput*-Lhr exhibits no preference for DNA:DNA or DNA:RNA duplex [7], *Msme*-Lhr prefers to unwind DNA:RNA duplexes in which the displaced strand is RNA [6]. Finally, the importance of *E*co*l*-Lhr rose from its occurrence in a cluster with the gene encoding RNase T, a ribonuclease involved in the maturation of stable RNAs, as well as in DNA repair pathways [8]. This interaction occurs at the transcriptional level, as the Lhr and RNase T are co-transcribed, but no interaction at the protein level was reported yet either in vitro or in vivo.

In Archaea, some genome annotations record two types of Lhr proteins, called here aLhr1 and aLhr2. The aLhr1 and aLhr2 exhibit a “Lhr-Core” domain organization [14]. aLhr1 has an additional cysteine-rich motif at its C-terminal end. Only a few Lhr from Sulfolobales (TACK) and Methanobacteriales (Euryarchaea) have been studied [15,16]. Lhr of *Sulfolobus islandicus* (SiRe_1605) was found to be important for the transcription of genes in nucleotide metabolism and DNA repair [17]. Monomeric Lhr of *Sulfolobus solfataricus*, also known as Hel112, was characterized in vitro as an ATP-dependent DNA helicase with a 3′-5′ polarity and a preference for forked DNA substrates [16]. Lhr of *Sulfolobus acidocaldarius* (saci_1500, also named RecQ-like helicase) was found to be important for DNA repair after UV-induced stress [18]. In vitro, Lhr of *Methanothermobacter thermautotrophicus* (*Mthe*) was also found to have 3′-5′ directional DNA translocase activity and to act on forked DNA structures. In a genetic assay, its expression gave a phenotype identical to the DNA helicases Hel308 and RecQ involved in replication-coupled DNA repair [15]. In addition, aLhr2 of *Pyrococcus abyssi* (*Paby*) was detected in the interaction network of proteins implicated in DNA replication and repair [19]. Recently, we also spotted *Paby*-aLhr2 as a partner of players in RNA metabolism. Indeed, *Paby*-aLhr2 was identified in the interaction network of the RNA helicase ASH-Ski2 together with the 5′-3′ and 3′-5′ RNA degradation machineries, aRNase J and the RNA exosome, respectively [20] (Appendix A). It should be noted that DNA metabolism enzymes were also identified in this network. This raises questions about the role of the aLhr2 proteins in Thermococcales.

In this study, we highlighted archaeal Lhr-type proteins as ubiquitous enzymes by revisiting the Lhr-type proteins landscape, using in-depth phylogenomic analyses. We identified six distinct phylogenetic groups of Lhr proteins, three in Archaea and three in Bacteria. We also defined the phylogenetic groups to which each of the experimentally studied Lhr helicases belong to. To go further in understanding the relevance of the archaeal aLhr2 group members in DNA and/or RNA metabolism, we characterized the enzymatic properties of aLhr2 from the Thermococcales *Thermococcus barophilus* (*Tbar*-aLhr2). Our results allowed us to propose that *Tbar*-aLhr2 is a DNA/RNA helicase with significant annealing activity that acts on DNA:RNA hybrids and on RNA:RNA duplexes.

## 2. Materials & Methods

### 2.1. Building Lhr-Type Dataset

Completely sequenced genomes of 286 Archaea and 3769 Bacteria showing a high level of annotation were downloaded from EBI (http://www.ebi.ac.uk/genomes/; accessed on 3 May 2019). The complete genomes of these 4055 strains, their proteomes and EMBL features were managed with an in-house MySQL database. Moreover, we had previously performed the annotation of the protein sequences of these genomes against the conserved domain database downloaded from the NCBI (https://www.ncbi.nlm.nih.gov/Structure/cdd/cdd.shtml; accessed on 2 April 2019) using the *rpsblast* program [21]. The *hmmscan* program [22] was used to annotate proteins with Pfam (32.0) domains. To avoid redundancies due to multiple repetitions of strains of the same species, we retained only one strain per species. Conversely, in order to obtain a better coverage of Archaea’s diversity, 75 Asgard proteomes were retrieved from UniProt, although they do not have the same sequencing and annotation quality as the other archaeal genomes (35 Lokiarchaeota, 27 Thorarchaeota, 12 Heimdallarchaeota and one Odinarchaeota). An initial sample of 1381 Lhr protein candidates were identified using the COG1201 (Lhr-like helicase) annotation performed by *rpsblast*. In order to eliminate the false positives while keeping the most divergent sequences, we set an e-value threshold ≤ 1 × e^−04^ associated with an alignment covering of at least 30% of the COG. 

To identify Lhr-like families, we performed all-against-all *blastp* comparisons of our initial set of proteins with default parameters, except for *-max_target_seqs* which was set to 1381 sequences. The results were filtered to retain only the best bi-directional hits between proteins of different species. Protein relationships were then converted into a graph in which the vertices represent protein sequences, and the edges represent their relationships [23]. The edges were weighted by the average pairwise -log_10_
*E*-value. The graph was further processed by a graph-partitioning approach based on the Markov Clustering algorithm (MCL, [24]). The inflate factor (IF) value is an important parameter of MCL as it regulates the cluster granularity. We tested several IF values (from 2 to 6) and a partitioning into 9 stable classes was observed starting from an IF ≥ 4. Classes 1 to 9 have sizes of 615, 344, 220, 106, 68, 22, 3, 1, and 1 sequences, respectively. A single protein from *Pseudomonas viridiflava* (*A0A1Y6JKR4_PSEVI*) was not classified and was discarded as a false positive since it shared only a small region of similarity with PF00271. 

In order to facilitate phylogenetic reconstructions while preserving the diversity of sequences in the original sample, we represented sequences with more than 70% identity by a single sequence, the medoid. To achieve this, the edges of the previous graph with an identity < 70% were removed. This pruned graph was further processed by MCL to identify groups of closely related sequences (identity ≥ 70). This identified 352 groups (including 27 Asgard groups) composed of a unique sequence and 111 groups composed of many closely related sequences. For each of the 111 groups, we computed the medoid, i.e., the sequence with the minimal average dissimilarity to all the other proteins in the group. We added the constraint that its length should be close to the median length of all sequences of the group. This resulted in a set of 463 proteins composed of 352 unique sequences and 111 medoids. Eight unique sequences were discarded as they did not have the PF00270 (DEAD) and/or the PF00271 (Helicase_C) domains. The aLhr1 and aLhr2 sequences of *T. barophilus* belong to two groups of closely related sequences. As a result of our sample size reduction process, these two sequences were not selected as medoid. The selected medoids were aLhr1 from *Thermococcus profundus* (TproA01.ASJ02541.1) and aLhr2 from *Methanocaldococcus jannaschii* (MjanA01. AAB98279.1). As aLhr2 from *T. barophilus* is the subject of this experimental study, the *Tbar*-aLhr1 (TbarA01.ADT83607.1) and *Tbar*-aLhr2 (TbarA01.ADT83510.1) protein sequences were added back to our sample. The tree in Figure 2 (Results section) shows that each sequence of *T. barophilus* has a direct common ancestor with their respective medoid. Our final sample contains 457 sequences (Appendix A).

Since the Sfth helicases appear to be the closest related family of Lhr helicases, both families sharing a common ancestor [14], we used Sfth sequences to root our Lhr family tree. To build the Sfth sample, we applied a similar strategy as the one described above for Lhr protein identification. Sfth protein candidates were identified using the COG1205 annotation performed by *rpsblast*. Proteins that did not possess the three expected domains (PF00270 DEAD; PF09369 DUF1998; PF00271 Helicase_C) were excluded. Using an identity threshold of 55%, we obtained 24 medoid sequences that we further used as representatives of this family (Appendix A). 

### 2.2. Alignment of the Core Helicase Domain

In order to eliminate the variability of the N- and C-term regions of the proteins, we extracted the central domain of the SF2 helicase core, i.e., the RecA1 and RecA2 regions (Figure 1). The coordinates of the alignment of the sequences with the PF00270 and PF00271 domains were used to extract both regions, which were then merged for each sequence. These sequences were aligned with *mafft* [25] (parameters: -reorder -localpair -maxiterate 1000). In order to improve the quality of the alignments and to keep as much information as possible, we used the *divvier* method [26] with the option *-divvy* (full divvying) and *-mincol* 4. This strategy was applied to the dataset containing only Lhr sequences and to the dataset composed of Lhr and Sfth sequences. 

### 2.3. Protein Family and Archaeal Species Trees

The best-fit amino acid substitution model for the data was selected with *modeltest-ng* [27] and the phylogenetic trees were inferred using the *iq-tree* software [28]. The same best model was selected for both datasets (-m LG4M + I). Branch supports were measured with ultra-fast bootstrap approximations (*-bb* 1000) and single branch tests (*-alrt* 1000). The trees were annotated and visualized with the online tool Interactive Tree Of Life (iTOLv5, https://itol.embl.de; accessed on 3 May 2019) [29].

To construct the archaeal species tree, we used the 122 markers that have been identified as reliable for phylogenetic inference [30]. We first thought of including the Asgard species in the tree but as their genomes are mostly partial, too many markers were missing for this to be feasible. Therefore, the tree was built by taking into account 219 archaeal species. The set of 122 protein markers was characterized by HMM profiles from the Pfam (v27) and the TIGRFAMs (v15.0) databases. For each genome included in this study, the proteins were identified by using each Pfam entry as query in the *hmmsearch* program from the HMMER 3.3.1 package (downloaded from http://hmmer.org/; accessed on 3 May 2019 [22]) with the *--cut_tc* (trusted cutoff) parameter. The *hmmsearch* output domain file was parsed to extract, for each genome and for each HMM profile, the best protein hit. Protein alignments with HMM profiles were merged for each marker. We thus obtained 122 sequence alignments. The columns of the alignments that had a high deletion frequency were removed with *trimal* (*-gt* 0.1) [31]. The quality of the alignments was estimated using the *t-coffee* transitive consistency score (TCS) [32]. The analysis of the results obtained on each alignment allowed us to (i) eliminate sequences with outlier TCS values and (ii) discard two alignments (PF04104.9 and PF01990.12) with a low overall TCS value (TCS < 65). The resulting 120 marker alignments were concatenated, the tree was inferred with *fasttree* [33] under the LG + GAMMA model, and branch support values were determined using 100 non-parametric bootstrap replicates. The tree was rooted on DPANN Archaea according to [34].

### 2.4. Genomic Context Analysis

We extracted the proteins encoded by the genes located less than 4000 bp upstream and downstream from the predicted *alhr2* genes. To obtain a functional characterization and classification of these proteins, they were annotated by *hmmscan* with TIGR HMM profiles. iTOL was used to associate these gene neighbourhoods to the species tree (DATASET_DOMAINS option). 

### 2.5. Expression Vectors

The Appendix A summarizes the oligonucleotides used in this study. All constructions were obtained by assembling PCR fragments using InFusion^®^ cloning kit (Takara). Using an appropriate set of oligonucleotides, the pET11b (untagged protein) vector was linearized by PCR amplification with the PrimeSTAR Max DNA polymerase (Takara), and the coding sequences of *T. barophilus* aLhr2 (TERMP_00533) and *P. abyssi* Hel308 (PAB_0592) were amplified from genomic DNA with the Phusion High-Fidelity DNA polymerase (ThermoFisherScientific). The pET11b vectors expressing the aLhr2-T215A, aLhr2-W577A and aLhr2-I512A variants were generated by site-directed mutagenesis of their wild-type counterpart with appropriate sets of oligonucleotides using the QuikChange II XL Kit (Stratagene). The pET11b vectors expressing the truncated aLhr2-ΔDom4 and the Domain 4 by itself (aLhr2-Dom4) were constructed by reverse PCR on the pET11b-aLhr2-WT using specific phosphorylated oligonucleotides and by DNA ligation (T4 DNA ligase). 

### 2.6. Purification of Tbar-aLhr2 Recombinant Proteins

*E. coli* BL21-CodonPlus (DE3) cells freshly transformed with pET11b-aLhr2, pET11b-aLhr2-T215A, pET11b-aLhr2-W577A, pET11b-aLhr2-I512A, pET11b-aLhr2-ΔDom4 and pET11b-aLhr2-Dom4 vectors were grown in 400 mL of LB medium at 37 °C. Protein production was induced at OD_600 nm_ 0.8 with 0.2 mM IPTG. After 3 h of induction at 30 °C, the cells were collected, suspended in 10 mL of lysis buffer (50 mM Tris-HCl pH 7.5, 150 mM NaCl, 10% glycerol) supplemented with 1 mg·mL^−1^ of lysozyme and a mix of EDTA-free protease inhibitor (cOmplete^TM^, Roche, Merck KGaA, Darmstadt, Germany), and lysed by sonication (4 × [5 × 10 s], 50% cycle, VibraCell Biolock Scientific). The cleared extracts, obtained by centrifuging the crude extracts (20,000× g, 4 °C, 20 min), were treated with a mix of RNase A (20 µg·mL^−1^), RNase T1 (1 U·µL^−1^) and DNase I (20 µg·mL^−1^) containing 10 mM of MgCl_2_ for 30 min at 37 °C. After a heating step at 70 °C for 20 min, the extracts were further clarified by centrifugation (20,000× g, 4 °C, 20 min). First, the recombinant proteins were purified from the soluble fractions to near homogeneity using FPLC (Fast Protein Liquid Chromatography, Äkta-purifier10, GE-Healthcare) and specific columns (GE Healthcare): for wild type aLhr2, the punctual mutants, and aLhr2-Dom4, by a cation exchange chromatography (Hitrap SP HP); for aLhr2-ΔDom4 by a heparin column (Heparin FF) with a linear gradient of NaCl (300 mM to 1 M). Then, all recombinant proteins were loaded on a size-exclusion HiLoad 16/60 Superdex 200 PG column in 20 mM HEPES pH 7.5, 300 mM NaCl, 10% glycerol buffer.

### 2.7. Preparation of Radiolabelled Nucleic Acid Substrate

The 26-nt RNA (RNA_26_) and all the DNA (DNA_26_, DNA_31_, DNA_50_ and DNA_59_) oligonucleotides were synthesized by Eurofins. The 50-nt RNA substrate (RNA_50_) was obtained by in vitro transcription from a PCR fragment where DNA_50_ was fused to the T7 promoter using the MEGAscript kit (Ambion). The DNA and RNA substrates were 5′-end radiolabelled using T4 polynucleotide kinase and γ-^32^P-ATP. To prepare nucleic acid duplexes, the short DNA or RNA oligonucleotide was radiolabelled, mixed with an unlabelled DNA or RNA complementary strand at a 1:1 molar ratio (100 nM each), incubated for 5 min at 95 °C in 1X SSC buffer, and then slowly cooled at room temperature. The nucleotide sequences of all the substrates used in this study are given in Appendix A.

### 2.8. ATPase Hydrolysis Assay

500 nM of recombinant protein were mixed with 5 nM of DNA_50_ or DNA_59_:DNA_31_ substrates in a 50 mM Hepes pH 7.5, 50 mM KCl, 5 mM MgCl_2_, 2 mM DTT buffer and preincubated for 10 min at 65 °C. 2 mM ATP and 0.85 µCi γ-32P-ATP were added at the 0 time point. The kinetic process was performed at 65 °C. At the indicated time, aliquots were spotted directly onto the TLC plate (PEI-cellulose, Macherey Nagel SAS, Hoerdt, France). TLC were developed with 0.25 M KH_2_PO_4_. Radioactive signals were measured using a PhosphorImager device (Typhoon Trio) and quantified with MultiGauge software (FujiFilm). The percentage of ATP versus ADP was plotted over time. Identical experiments were performed with 5 nM of DNA_50_ or RNA_50_ with a range of ATP concentration (0.025, 0.05, 0.1, 1, and 2 mM) in triplicates. The plots were derived using GraphPad Prism 7 software.

### 2.9. Nucleic Acid Binding Assay

Double filtration binding assays were performed with range of protein concentrations from 0 to 350 nM and 0.5 nM of ^32^P-labelled RNA or DNA substrate using a Slot blot device (Amersham Biosciences). The protein was preincubated for 10 min at 65 °C in 25 mM Tris-HCl pH 8, 50 mM NaAc, 5 mM MgCl_2_, 2.5 mM β-Mercaptoethanol. After adding the substrate, the reactions were incubated for 15 min at 30 °C. Free nucleic acids were separated from nucleoprotein complexes on double filtration systems using Nylon and Nitrocellulose membranes (Amersham^TM^ Hydond-N and Protran, respectively). Radioactive signals were measured using a PhosphorImager device and quantified with MultiGauge software. The apparent dissociation constants K_D_ were calculated using GraphPad Prism 7 software. 

The oligomerization state of *Tbar*-aLhr2 was determined by size exclusion chromatography. After cation exchange chromatography (see Section 2.6), the protein was concentrated and desalted using a Vivaspin centrifugal concentrator with a molecular weight cut-off of 50,000 Da (Sartorius). About 2 µM of protein was preincubated at 65 °C for 10 min in 25 mM Tris-HCl pH 7.5, 50 mM NaAc, 5 mM MgCl2, 300 mM NaCl. After adding 1 µM of DNA_50_ substrate, the reaction was incubated for 15 min at 30 °C, and the mixture was loaded on a size-exclusion Superdex 200 Increase 10/300 GL column in 25 mM Tris-HCl pH 7.5, 50 mM NaAc, 5 mM MgCl2, 300 mM NaCl, 10% glycerol buffer. The fractions were analysed by Coomassie-blue SDS-PAGE and Western blotting.

### 2.10. Helicase (Unwinding) Assay

The unwinding assays were done with 250 nM of protein, 5 nM of α-^32^P-labeled nucleic acid duplex and a 200-fold excess of the unlabelled oligo trap (1 µM). The protein was preincubated separately for 5 min at 65 °C in 25 mM Tris-HCl pH 8, 50 mM NaAc, 2.5 mM β-Mercaptoethanol, 5 mM MgCl_2_, 25 mM ATP. After addition of the recombinant protein (250 nM), the reaction mixtures were incubated at 65 °C for the indicated times and then quenched with 0.5% SDS, 40 mM EDTA, 0.5 mg·mL^−1^ Proteinase K, 0.1% Bromophenol blue, and 20% glycerol. The reaction products were separated on a native 8% polyacrylamide gel (1X TBE, 0.1% SDS) by electrophoresis in 1X TBE (200Volts, 90 min). Radioactive signals were measured using a PhosphorImager device and quantified with MultiGauge software. All assays were repeated at least three times.

### 2.11. Strand-Annealing Assay

5 nM of radiolabelled substrates and 250 nM of recombinant protein were preincubated separately for 5 min at 65 °C in 25 mM Tris-HCl pH 8, 50 mM NaAc and 2.5 mM β-Mercaptoethanol. The reactions were started by mixing the protein and nucleic acid samples. After incubation at 65 °C, samples of 5 µL were withdrawn at the indicated time points. The reactions were quenched and analysed as described in Section 2.10. All assays were independently repeated at least three times.

## 3. Results

### 3.1. Phylogenomic Studies of Lhr-Type Helicases in Archaea & Bacteria

Our initial Lhr library was composed of 1380 proteins that were identified by a similarity search against the COG1201 profile of the COG database that covers the “Lhr core” organization of Lhr helicases, i.e., the two conserved RecA1 and RecA2 domains, the winged-helix motif and the Domain 4 (Figure 1). To explore the family organization, the protein relationships were converted into a graph that was further processed with MCL to identify groups of Lhr proteins. Nine groups were obtained including five main classes of 615, 344, 220, 106, 68 sequences respectively. The aLhr2 sequences from *P. abyssi* (SP: Q9UZM4) and *T. barophilus* (SP: F0LJX3, TbarA01.ADT83510.1) belonged to the MCL class 1 as well as the characterized Lhr from *S. solfataricus* (SP: P95949, SSO0112), *S. acidocaldarius* (SP: Q4J8R1, saci_1500) and *M. thermautotrophicus* (SP: O27830, MTH_1802). The aLhr1 sequences from *P. abyssi* (SP: Q9V0H2) and *T. barophilus* (SP: F0LKE9, TbarA01.ADT83607.1) as well as the studied sequence of *S. islandicus* (SiRe_1605) were found in the MCL class 3 (Appendix A). In our sample, *S. islandicus* is represented by the strain L.S.2.15 (SislA01.ACP36165.1, SP: C3MR20) whereas the REY15A strain is the one that has been functionally studied; aLhr1 from L.S.2.15 presents 99.8% of identity with its REY15A ortholog. For the bacterial Lhr helicases, the proteins from *M. smegmatis* (SP: A0QT91) and *E. coli* (SP: P30015) have been found in MCL class 1, while the one of *P. putida* (SP: Q88NV1, PP_1103) belongs to MCL class 2 (Appendix A).

To go further, we computed two phylogenetic trees. To avoid bias due to the overrepresentation of closely related species in public databases, sequences showing more than 70% identity were displayed by a representative sequence (see Material and Methods). This facilitated phylogenetic reconstructions while preserving the diversity of sequences in the original sample. Our reduced sample contained 457 proteins. In order to eliminate the variability of the C-terminal regions of the Lhr proteins, and to allow the comparison with the Sfth helicases, multiple alignments were performed with the SF2 helicase core composed of the RecA1/RecA2 domains. The first Lhr family tree was rooted by adding 24 reference sequences of the Sfth helicase family (Appendix A) that appears to be the closest related family of the Lhr helicases [14]. Then, we constructed a second tree on the sole Lhr sequences that were rooted by using the most external Lhr subtree identified above (Figure 2). The topologies of the trees were consistent with the MCL classes obtained on complete sequences, with six subtrees clearly identified. 

The first colour-coded ring around the trees indicates the bacterial (purple), the archaeal (green) and the Asgard (yellow) genomes, respectively. The second colour-coded ring figures the MCL groups as stated in the figure legend. The colour-coded pie slices indicate the boundaries of each subtree. Two subtrees, one corresponding to MCL class 3 and the other, smaller, corresponding to MCL class 6, contain only archaeal sequences. A third subtree corresponding to MCL class 2 encloses only bacterial sequences. Based on the tree topologies, the MCL class 1 can be clearly subdivided into two subtrees, one containing only bacterial sequences and the other only archaeal sequences. Finally, the last subtree regroups the sequences belonging to the remaining MCL classes. Except for the MCL classes 7, 8 and 9 that contain only few sequences (one to three), we can notice that the MCL classes 4 and 5 correspond to groups of sequences that share a common ancestor. The location of the MCL class 2 subtree is different in the two trees. In the tree rooted with Sfth, it shares a common ancestor with the bacterial subtree of MCL class 1 (Appendix A) while in the tree based on Lhr sequences alone (Figure 2), it forms a group external to the MCL class 1 sequences that is under the same ancestor node. Since the branch shows a weak bootstrap support in the Sfth tree (SH-like approximate likelihood ratio test (alrt) < 60) (Appendix A), we favoured the topology obtained on the sole Lhr proteins (Figure 2). It can be noticed that the MCL class 2 subtree presents an acceleration of the rate of evolution that could be at the origin of the instability of its placement in the different trees.

The domain organization based on the Pfam profiles of each protein are shown as outer rings on the tree (Figure 2). While they all possess the RecA1 (PF00270 entry) and RecA2 (PF00271 entry) domains that form the helicase core, Domain 4 (PF08494 entry named “DEAD-associated domain”) is present in the subtree corresponding to MCL classes 1 to 3 and is missing from the subtree corresponding to MCL classes 4 to 9. These sequences have a shorter C-terminal region that is not characterized by any conserved domain except for class 6 sequences that have a highly deteriorated Domain 4. The bacterial MCL class 1 sequences have a longer C-terminal region containing an additional HTH_42 domain (PF06224 entry). This is also the case for some archaeal MCL class 1 sequences.

The groups of Lhr helicases that correspond to the different subtrees were named based on the MCL classes of the experimentally characterized Lhr proteins and on their domain organization. In Bacteria, we detected three orthologous groups of Lhr proteins that we referred to as: bLhr (MCL class 2) when they were restricted to the “Lhr-core”; bLhr-HTH (bacterial MCL class 1) based on the additional HTH_42 domain at their C-terminal end; and finally Lhr-like when their sequences did not possess Domain 4 (mostly MCL classes 4 and 5). This last group contained some scattered sequences of Archaea that where probably acquired by horizontal gene transfers. To our knowledge, it is the first time that three different groups of bacterial Lhr helicases are reported. In Archaea, we found the already reported aLhr1 (MCL class 3) and aLhr2 (archaeal MCL class 1) groups [14]. We also identified for the first time a third small group that we named aLhr3 (MCL class 6) and that is characterized by a highly deteriorated Domain 4. 

To go further, the taxonomic distribution of the archaeal Lhr groups was performed (Figure 3). Note that the Asgard with incomplete genomes have not been included in the species tree. aLhr1 members were found in two DPANN genomes out of six, in all TACK except in two Candidatus genomes (*C. nitrosmarinus catalina SPOT01* and *C. nitosopumilus sp. AR2*) for which no Lhr proteins were detected, and in almost all the Euryarchaeota genomes with the exception of the Methanopyraceae, the Methanococcales and the Methanobacteriales. Members of the aLhr2 group were found in the DPANN genomes, in most TACK genomes except in Thaumarchaeota and Thermoproteales and in the majority of Euryarcheota with the exception of the Methanomicrobiales and the Methanosarcinales. It can be noticed that in the genomes of Thermoproteales which do not contain the *alhr2* gene, two paralogous *alhr1* genes were found. Members of the aLhr3 group were only found in Sulfolobales, Desulfurococcales and Acidilobales. Interestingly, these genomes usually also encode a member of the aLhr1 and aLhr2 groups. Finally, two genomes *Candidatus Methanomassiliicoccus intestinalis Issoire-Mx1* (Methanomassiliicoccales) and *Aciduliprofundum sp. MAR08-339* (Aciduliprofundum) encode an Lhr-like protein in addition to aLhr1 and aLhr2.

As shown in Figure 2, some aLhr2 proteins are longer than the majority. We therefore analysed more closely their domain organization (Appendix A). aLhr2 proteins from Methanomassiliicoccales harbour a longer C-terminal region containing an additional HTH_42 domain (PF06224), also found in bLhr-HTH proteins. An extended C-terminal region was found in other genomes such as the Thermoplasmatales, but with no detected Pfam domain. Interestingly, in few cases, aLhr2 proteins have their RecA2 domain (like in *P. horikoshii* and *Methanocaldococcus sp FS406-22*) or RecA1 domain (like in *Natrialba magadii ATCC 43099*) split by an intein_splicing domain (PF14890) in which a LAGLIDADG domain (PF14528) is inserted.

Concerning the Asgard sequences, we identified 22 aLhr1 proteins and only five aLhr2 proteins distributed as follows (Figure 2). aLhr1 as encoded in 18 Thorarchaeota, in three Heimdallarchaeota and in the only representative of Odinarchaeota. aLhr2 was found in four Heimdallarchaeota and one Thorarchaeota. Since these genomes are incomplete, it is difficult to draw conclusions. However, none of the genomes analysed possessed two Lhr helicases, and in the only complete genome (*Prometheoarchaeum syntrophicum*), we did not identify any Lhr protein. Moreover, we observed that the Asgard aLhr1 proteins form a group that shares a common ancestor node within the aLhr1 subtree (Figure 2). On the other hand, the few Asgard aLhr2 sequences are scattered in the aLhr2 subtree, suggesting acquisition by horizontal gene transfer. However, more data are needed to confirm these hypotheses.

Finally, we analysed the genomic context of the genes encoding aLhr1 and aLhr2 in order to detect microsynteny since such conservation can give information about function and functional interactions. The products of the genes surrounding the *alhr1* and *alhr2* genes were functionally annotated with TIGR HMM profiles. For aLhr1 and aLhr2, no gene conservation was found across all the archaeal genomes studied (Appendix A). Since our experimental work concerns aLhr2 from *T. barophilus*, we looked at the *alhr2* gene neighbourhood more carefully. Conservation of two different neighbour genes was observed in genomes not closely related. The first one was found either upstream or downstream of the *alhr2* gene in DPANN genomes, in two Thermoproteales genomes and in some Euryarchaeota genomes scattered across the phylogeny (in green for the alhr2 context, Appendix A). Its gene product showed similarity with the TIGR0024 profile (putative phosphoesterase), the COG1407 (Predicted ICC-like phosphoesterase) and the cd07391 whose members include archaeal and bacterial proteins homologous to the *Pyrococcus furiosus* PF1019 protein. The domain present in these members belongs to the metallophosphatase (MPP) superfamily. One can notice that such a gene was also found upstream of the *alhr1* gene in six out of nine Methanomicrobiales genomes, in eight out of 20 Methanosarcinales genomes, in most genomes of Halobacteriales and Haloferacales and finally in two genomes of Natrialbales (in yellow on the *alhr1* context, Appendix A). However, its presence in the neighbourhood of the *alhr* genes appears to be mutually exclusive, either upstream of *alhr1* or in the neighbourhood of *alhr2*. Interestingly, in Bacteria, a strongly conserved homologous gene called MPE for metallophosphoesterase was also found in the vicinity of *blhr* genes (in 276 genomes out of 319) but not in the neighbourhood of *blhr-HTH* genes that showed no apparent conservation. This is in agreement with previous published results [7]. Finally, a second gene was found just upstream of the *alhr2* gene, either in the same or reverse orientation, in most genomes of Sulfolobales, Desulfurococcales and a group of genomes from Thermococcaceae (in yellow for *alhr2* context; Appendix A). Its gene product showed a low similarity with the TIGR03937 (poly-beta-1,6 N-acetyl-D-glucosamine synthase) and the COG1215 whose members are described as glycosyltransferases, and are probably involved in cell wall biogenesis. However, they exhibit very weak similarities that prevent us from making any functional prediction. 

### 3.2. The Biochemical Properties of aLhr2 of Thermococcus Barophilus

In view of previous studies identifying *Paby*-aLhr2 as part of the interaction networks of proteins involved in DNA and RNA transactions [17,18], we focused our attention on characterizing the biochemical properties of Thermococcales aLhr2. We initially chose to study the *P. abyssi* version of aLhr2. However, because the corresponding recombinant protein was toxic when expressed in *E. coli* cells, we decided to perform in vitro assays with aLhr2 from *T. barophilus* (*Tbar*-aLhr2). It should be noted that *P. abyssi* and *T. barophilus* are two closely phylogenetically related hyperthermophiles Euryarchaea from the order of Thermococcales that were both identified in the same ecological niche, deep-sea hydrothermal vents [35]. Since *Paby*-aLhr2 and *Tbar*-aLhr2 amino acid sequences share 90% of similarity, we chose to assume that the two orthologous proteins had the same biochemical properties. In addition, we previously showed that antibodies raised against *P. abyssi* proteins recognized their *T. barophilus* counterparts [20]. Interestingly, while the genome of *P. abyssi* cannot be modified with current techniques, *T. barophilus* is now amenable to genetic manipulations, such as gene deletion [18,33]. 

Using purified untagged *Tbar*-aLhr2 recombinant protein that we have shown to be monomeric (Appendix A), we performed a series of assays to determine properties inherent to helicase enzymes. In the following, we report the capacity of *Tbar*-aLhr2 to hydrolyse ATP, to bind nucleic acids, and to form and unwind duplexes. To do so, we designed a panel of basic RNA and DNA substrates (26 to 50 nucleotides long; Appendix A) with sequences based on the first study reporting the in vitro activity of the archaeal DNA helicase Hel308 of *M. thermautotrophicus* (*Mthe*) [35]. The in vitro assays were performed with the wild-type protein, with proteins harbouring substitutions in the signature motifs of aLhr2 proteins (T215A, I512A and W577A), and with proteins deleted of or restricted to Domain 4 (∆Dom4 and Dom4, respectively) (Figure 4 & Appendix A). The residue T215 of Motif III of the SF2 core was predicted to be important for coordination of ATP hydrolysis and nucleic acid binding [36]. The residue W577 of the Domain 4 was highly conserved and was shown to be important in the coupling of ATP hydrolysis and DNA translocation in *M. smegmatis* [7,9].

#### 3.2.1. Tbar-aLhr2 Is a Nucleic-Acid Dependent ATPase

To measure the capacity of *Tbar*-aLhr2 to hydrolyse ATP, we performed an ATPase assay with an excess of ATP, and in the absence or presence of a single-stranded DNA_50_ molecule or a DNA_59_:DNA_31_ hybrid (Appendix A). The release of inorganic phosphate was followed over time. Our results show that both DNA_50_ and DNA_59_:DNA_31_ stimulate the *Tbar*-aLhr2 ATPase activity (Appendix A). In absence of nucleic acids, only residual ATP hydrolysis was observed. Altogether, these results show that *Tbar*-aLhr2 is a nucleic acid-dependent ATPase. As observed for other SF2 helicases, ATP hydrolysis can be inactivated by mutating the active site formed by the two RecA1 and RecA2 domains [37,38]. The *Tbar*-aLhr2-T215A protein mutated in the conserved motif III had only a residual ATPase activity in presence of DNA_50_ (Appendix A). Alone, *Tbar*-aLhr2-Dom4 exhibited no ATPase activity. On the other hand, the truncated *Tbar*-aLhr2-ΔDom4, restricted to the RecA1/RecA2 domains and the WH motif, conserved its capacity to hydrolyse ATP but with less efficiency (Appendix A). Altogether, these results suggest that Domain 4 is critical for the optimal ATPase activity of *Tbar*-aLhr2.

Moreover, by assessing ATPase activity rate, we showed that *Tbar*-aLhr2 has no apparent preference for DNA or RNA molecules (Figure 5). To do so, the kinetics of ATP hydrolysis were measured at different concentrations of ATP in the presence of 50-nt long DNA_50_ or RNA_50_ molecules (Appendix A). The calculated ATPase rates of *Tbar*-aLhr2 were identical. We also confirmed that the ATPase rates of the T215A and ∆Dom4 mutants were greatly affected (Figure 5; Appendix A).

#### 3.2.2. Tbar-aLhr2 Is a Nucleic Acid Binding Protein with Similar Affinities for RNA and DNA Molecules

We used a nitrocellulose-filter binding assay to test the capacity of wild type *Tbar*-aLhr2 to bind single-stranded nucleic acid molecules (Figure 6A, left panel) or homoduplex substrates (Appendix A). Briefly, an increased concentration of protein was incubated with 5 nM of substrates (Appendix A). The nucleoprotein complexes were separated from free nucleic acids by double filtration on nitrocellulose and nylon membranes. The percentage of bound fraction retained by the nitrocellulose membrane was plotted against the protein concentration (Figure 6A, left panel). The binding curves show a sigmoidal shape with a Hill coefficient superior to 1 (S-shape curves), indicating positive binding cooperativity or multiple binding sites. Most likely, more than one molecule of protein binds to multiple sites on a single molecule of nucleic acids. This was confirmed by size exclusion chromatography performed in presence of DNA_50_ (Appendix A). Alone, *Tbar*-aLhr2 mainly eluates as a monomer. When pre-incubated with DNA_50_, *Tbar*-aLhr2 also assembles as a multimer. Based on column calibration, the multimer seems to be constituted of four to five molecules of *Tbar*-aLhr2. 

The apparent dissociation constants Kd determined for each complex show that *Tbar*-aLhr2-WT binds single-stranded RNA and DNA substrates with the same affinity. A slightly higher affinity was observed for 50nt-long substrates when compared to their 26nt-long counterparts, but with less than 2-fold differences in Kd values it is not significant (Figure 6A, left panel; Table 1). Comparable affinities are also observed for RNA_50_:RNA_26_ and DNA_59_:DNA_31_ homoduplexes (Appendix A).

The *Tbar*-aLhr2-ΔDom4 mutant has similar affinities for 50nt-long DNA and RNA substrates as *Tbar*-aLhr2-WT, but has a lower affinity for the shorter 26nt-long substrates (Figure 6A, right panel; Table 1). While the difference in K_d_ values was 2.2-fold for the RNA substrates, the difference in Kd values for the DNA substrates was almost 4-fold. Moreover, we observed that Domain 4 of *Tbar*-aLhr2 by itself had the capacity to bind nucleic acids with a lower affinity (Figure 6B; Table 1). Altogether, these results suggest that Domain 4 significantly enhances the capacity of *Tbar*-aLhr2 to bind small nucleic acid substrates. This is consistent with the structure of *Msme*-Lhr (restricted to the first 1-856aa) in complex with AMP-PNP and a single-stranded 16-mer DNA molecule (PDB: 5V9X) showing the RecA2 and Domain 4 forming a clamp around the ssDNA [9]. 

#### 3.2.3. Tbar-aLhr2 Displaces Single-Stranded RNA from RNA:DNA or RNA:RNA Hybrids with a Preferred 3′ to 5′ Unfolding Directionality

To assess the helicase activity of *Tbar*-aLhr2, we tested its capacity to unwind nucleic acid duplexes. The assays were performed with homo- (DNA:DNA or RNA:RNA) or hetero- (DNA:RNA) duplexes with 5′ or 3′ overhangs (Appendix A). Note that to obtain stable DNA homoduplexes in our experimental conditions, we used longer ssDNA molecules (DNA_59_:DNA_31_) than for the heteroduplexes (DNA_50_:RNA_26_) or RNA homoduplexes (RNA_50_:RNA_26_). The shorter strand was radiolabelled at its 5′ end. Duplex unwinding was followed over time at 65 °C in the presence of 250 nM of *Tbar*-aLhr2. Unlabelled Trap oligo was added in excess to prevent new rounds of duplex formation. A protein-free control experiment was done to assess temperature-dependent unwinding (Appendix A).

The percentage of newly-formed single strands was plotted over time (Figure 7A, left panel). It took 90 min for wild type *Tbar*-aLhr2 to unwind more than 60% of 3′ overhang 3′RNA_50_:RNA_26_ and 3′DNA_50_:RNA_26_ duplexes. These unwinding activities, that were rather slow when compared to other helicases, were even less efficient for a 5′ overhang 5′RNA_50_:RNA_26_ duplex (Figure 7A, left panel). This indicates that *Tbar*-aLhr2 has a slow helicase activity and a preference for 3′ overhang duplexes. In addition, we observed that *Tbar*-aLhr2 was not able to unwind a 3′ overhang 3′DNA_59_:DNA_31_ duplex. Indeed, almost no unwinding as observed for the control without protein (Figure 7A, left panel). This suggests that *Tbar*-aLhr2 displaces only ssRNA molecules and not ssDNA from homo- or hetero-duplexes. 

To go further in understanding the unwinding activity of *Tbar*-aLhr2, we compared it to that of *Paby*-Hel308, which was reported to be a helicase with DNA unwinding activity [39]. Conversely to *Tbar*-aLhr2, *Paby*-Hel308 was able to rapidly unwind 80% of 3′ overhang 3′DNA_59_:DNA_31_ duplexes after only 2 min of incubation (Figure 7A, right panel). While both proteins have similar binding affinities for nucleic acids and do not show any specificity for DNA_50_ substrates (Table 1), their activities differ because of their respective substrates and unwinding velocities. Therefore, Hel308 and aLhr2 that were previously reported to be involved in DNA repair [15,40] most likely operate to perform different tasks in DNA transactions.

All the previous experiments were performed in presence of ATP. We also performed unwinding experiments of 3′ overhang 3′RNA_50_:RNA_26_ duplexes in the absence of ATP or in the presence of non-hydrolysable ATP analogues. Unexpectedly, we observed that in the absence of ATP, the unwinding activity was comparable to that obtained in the presence of ATP (Appendix A). In our experimental conditions, ATP binding and hydrolysis seem to not be required for slow-rate unwinding activity. It is possible that the energy required for duplex separation is provided by the binding of the protein to the nucleic acid substrate. Moreover, the addition of AMP-PNP or ATPγS abolished completely the observed unwinding activity (Appendix A). The binding of ATP analogues might somehow trap *Tbar*-aLhr2 in an inactive state. 

#### 3.2.4. Tbar-aLhr2 Forms 3′ Overhang Duplexes with No Preference for RNA or DNA Molecules

To test the capacity of wild type *Tbar*-aLhr2 to anneal nucleic acid strands, we did the reverse experiment and followed the formation of homo- and hetero-duplexes from complementary single-stranded DNA or RNA substrates (Appendix A). Strand annealing was followed by treatment at 65 °C in the presence of 250 nM of *Tbar*-aLhr2 (Appendix A). The percentage of newly formed duplexes was plotted over time (Figure 7B). A protein-free control experiment was conducted to assess temperature dependent annealing (Appendix A). In the absence of ATP, *Tbar*-aLhr2 was able to rapidly anneal nucleic acid single strands to form 3′ overhang 3′DNA_50_:RNA_26_, 3′RNA_50_:RNA_26_ and 3′DNA_50_:DNA_26_ duplexes with no major differences (Figure 7B, left panel). After 10 min of reaction, duplex formation plateaued at 60%, 70% and 75%, respectively. On the other hand, the annealing velocity was drastically reduced when the single strands formed 5′ overhang 5′RNA_50_:RNA_26_ duplexes. At 10 min, almost no duplexes were formed in the control without protein. It seems that in our experimental conditions, *Tbar*-aLhr2 mainly adopts an annealing-competent state. 

Again, we compared the activity of *Tbar*-aLhr2 and *Paby*-Hel308. In the same experimental conditions, we showed that *Paby*-Hel308 was unable to rapidly form 3′DNA_50_:DNA_26_ duplexes (Figure 7B, right panel). This result is consistent with Hjm/Hel308 from *Sulfolobus tokodaii* that only exhibits structure specific ssDNA annealing [39]. This also confirms that in vitro, characteristics of *Tbar*-aLhr2 and *Paby*-Hel308 differ with a distinct range of activities.

In the presence of ATP, *Tbar*-aLhr2 annealing capacity was reduced with only 20% of 3′RNA_50_:RNA_26_ duplexes formed after 10 min (Appendix A). It is possible that the protein conformation switched to an inactive state upon the binding of ATP. This effect was even more drastic in presence of non-hydrolysable ATP analogues (AMP-PNP and ATPγS) (Appendix A). While this could mean that ATP hydrolysis switches *Tbar*-aLhr2 in an unwinding conformation, it seems unlikely since ATP does not seem to stimulate its unwinding activity, which is rather slow (Appendix A). It is more likely that the annealing competent state of *Tbar*-aLhr2 is sensitive to the presence of ATP or non-hydrolysable ATP analogues.

#### 3.2.5. Domain 4 Is Essential for the Unwinding and Annealing Activities of Tbar-aLhr2

We revealed that Domain 4 stimulates the ATPase activity of *Tbar*-aLhr2 and by itself has the capacity to bind nucleic acids (Figure 5 and Figure 6). Here, we investigated its role in terms of unwinding and strand-annealing activities (Figure 8A,B, respectively). The protein *Tbar*-aLhr2-ΔDom4 which is deprived of Domain 4 had only residual activities. Consistently, *Tbar*-aLhr2-W577A, in which the highly conserved tryptophan at position 577 of Domain 4 was mutated (Figure 4), was also defective for both reactions (Figure 8). We concluded that Domain 4 is essential for the formation of either an active unwinding or annealing-competent state of *Tbar*-aLhr2. The protein variant *Tbar*-aLhr2-I512A with a mutated residue in the WH domain showed similar activities as the wild-type suggesting that this highly conserved residue is not important for in vitro helicase activity in these experimental conditions (Figure 8).

## 4. Discussion

Helicases are key enzymes involved in processes that depend on DNA and RNA transactions. They are known to use the energy of ATP to unwind nucleic acids and remodel protein-nucleic acid complexes. Here, we focused on the Thermococcales SF2 helicase aLhr2. *P. abyssi* aLhr2 (*Paby*-aLhr2) was found in the network of proteins involved in DNA replication and repair [19]. However, our recent observation also found that *Paby*-aLhr2 was part of the interaction network of proteins involved in RNA processing [20]. This questions the function(s) of Thermococcales aLhr2. Since *Paby*-aLhr2 is toxic when expressed in *E. coli*, we investigated the in vitro activity of its orthologue in *T. barophilus* (*Tbar*-aLhr2). We determined that it is a monomeric DNA/RNA helicase able to process DNA:RNA and RNA:RNA duplexes. Moreover, for other archaeal aLhr helicases that were proposed to be DNA helicases involved in DNA repair and recombination [15,41], it was unclear if they belonged to the same aLhr2 group. We also interrogated the relationship that exists between the archaeal and bacterial Lhr proteins. Thus, we performed extensive phylogenomic analyses to elucidate the evolutionary links between Lhr proteins in Archaea and in Bacteria.

The Lhr-type proteins are defined by their unique domain organization [9]. The “Lhr core” is composed of a SF2 helicase core, a winged-helix motif and an Lhr-specific Domain 4 (Figure 1). After recovering and annotating the archaeal and bacterial Lhr-type proteins based on domain organization, we established their partition in MCL groups and computed a protein family tree with the conserved SF2 core region (Figure 2). We could distinguish six groups sharing a common origin: three groups include only Lhr proteins from Archaea: aLhr1, aLhr2 and aLhr3; two groups include only Lhr proteins from Bacteria: bLhr and bLhr-HTH; and the last group, while dominated by Bacteria, includes few archaeal proteins (Lhr-like). The few archaeal sequences belonging to the Lhr-like group are scattered on the trees and should have been acquired from Bacteria through horizontal gene transfers. The Lhr groups were named based on previous studies [14] and domain organization. According to the tree topology (Figure 2), the sequences of the bLhr-HTH and aLhr2 groups share a common ancestor that predates the divergence of Bacteria and Archaea, suggesting that these sequences are orthologous and may have conserved similar roles in these genomes. The sequences of the bLhr group are characterized by significantly longer branches than those of the other groups. This reflects an acceleration in their rate of evolution and could be responsible for the instability of the anchoring of this group between the Sfth rooted (Appendix A) and Lhr-like rooted (Figure 2) trees. Despite this difference in localization, the bLhr sequences share, on both trees, a common hypothetical ancestor with the bLhr-HTH and aLhr2 helicases. On the other hand, the aLhr1 group does not appear directly related to bacterial sequences. 

While initial work identified two groups of archaeal Lhr proteins, aLhr1 and aLhr2 [14], we identified a third group aLhr3 that has a highly deteriorated Domain 4 and that seems to be limited to the Sulfolobales, Desulfurococcales and Acidobales. On the other hand, aLhr1 and aLhr2 are widespread in archaeal genomes and often present together. Their absence in some genomes would most likely result from different independent events of gene loss. Interestingly, only four out of 219 have lost both genes. We defined the characterized Lhr proteins of *S. solfataricus*, *S. acidocaldarius*, and *M. thermautotrophicus* [15,16] to belong to the aLhr2 group. To date, no aLhr1 have been characterized. Genetic studies on a strain of *S. islandicus* deleted the gene encoding aLhr1 [17] suggesting that *Sisl*-aLhr1 has a role in DNA repair, as shown for *Ssol*-, *Saci*- and *Mthe*-aLhr2, but its biochemical properties are unknown. More work is needed to identify the common and/or specific functions of aLhr1 and aLhr2 in archaeal cells. In particular, since aLhr1 differs from aLhr2 by an additional cysteine-rich motif at its C-terminal end, the presence or not of an additional putative Zinc-finger might differentiate the proteins’ activities and partners. 

Interestingly, in eleven genomes the RecA2 (10/11) or RecA1 (1/11) domain of aLhr2 is spliced by an intein_splicing domain. Proteins containing this integration are dispersed on the tree suggesting independent acquisitions. Interestingly, ATPase domains were shown to be hot-spots for inteins integration, with 70% of all inteins residing in ATPase-containing proteins, and at many different integration sites [42]. While they are generally considered as being selfish parasites, intein splicing has recently been shown to be regulated by external stimuli such as temperature, pH, salt and DNA damage [42]. Thus, some aLhr2 proteins might be regulated at the post-translational level and activated upon stress.

In Bacteria, for the first time, we identified three groups of Lhr proteins: bLhr, bLhr-HTH and Lhr-like. While bLhr proteins are restricted to the “Lhr core”, the group of bLhr-HTH has an additional HTH_42 domain at its C-terminal end. As before, the presence of an extra domain interrogates its role in the protein’s activities and interaction with partners. To date, one bLhr from *P. putida* and two bLhr-HTH from *M. smegmatis* and *E. coli* have been studied, but the role of the HTH_42 domain has not been investigated. Interestingly, some aLhr2 from Methanomassilicoccales also possess an additional HTH_42 extension. Finally, the Lhr-like were defined as Lhr-type proteins, but while they possess a C-terminal region, no Domain 4 could be defined. Nonetheless, prediction of the structure of Lhr-likes from *Streptomyces coelicolor* showed that ttheir C-terminal domain adopts a structure that is similar to the structure of Domain 4 of *Msme*-bLhr-HTH (Appendix A).

In this study, we also report the in vitro activities of aLhr2 from *T. barophilus*. First, we showed that the ATPase activity of *Tbar*-aLhr2 is consistent with that measured for the bacterial *Msme*-bLhr-HTH and *Pput*-bLhr proteins [6,43]. We showed that *Tbar*-aLhr2 is a nucleic acid-dependent ATPase with no apparent preference for DNA or RNA molecules (Figure 5). Only the archaeal *Ssol*-aLhr2 was shown to be able to hydrolyse ATP in the absence of nucleic acids [41]. *Ssol*-aLhr2 also differs from the monomeric *Tbar*-aLhr2 and bacterial Lhr proteins by its low affinity for single-stranded DNA and by its oligomeric state that can be both monomeric and dimeric; monomers and dimers having specific biochemical activities. 

We also characterized *Tbar*-aLhr2 as a monomeric DNA/RNA helicase able to process DNA:RNA and RNA:RNA duplexes (Appendix A & Figure 7, left panels). Interestingly, we highlighted that the in vitro unwinding and annealing activities of *Tbar*-aLhr2 differ drastically from those of Hel308, described as a DNA helicase involved in DNA repair [35,44]. Indeed, we showed that while *Tbar*-aLhr2 is more prone to anneal nucleic strands than to unwind them, *Paby*-Hel308 unwinds 3′DNA:DNA homoduplexes but does not form them from ssDNA molecules (Figure 7). Altogether, these results clearly indicate that while both proteins are proposed as being involved in DNA repairs [15,40], they most likely perform different tasks in DNA transactions.

Among Lhr proteins, the capacity to process DNA:RNA hybrids is not specific to *Tbar*-aLhr2. Indeed, while the bacterial *Msme*-bLhr-HTH and *Pput*-bLhr, and the archaeal *Mthe*-aLhr2 were shown to process other substrates (forked DNA or Holliday junctions), they are also able to unwind DNA:RNA hybrids [6,7,9,15]. *Msme*-bLhr-HTH even prefers 3′-tailed RNA:DNA hybrids over DNA:DNA duplexes and was described as an RNA/DNA helicase [9]. The capacity of *Tbar-*aLhr2 to unwind more efficiently in vitro hybrids with a 3′ overhang strand that is indicative of a 3′ to 5′ polarity is also consistent with the polarity previously observed for its archaeal and bacterial counterparts [6,7,9,15,16].

*Tbar*-aLhr2 has a significant ability to anneal single-stranded nucleic acid substrates to form DNA:DNA, RNA:RNA or RNA:DNA duplexes with no apparent preferences (Figure 7). Both the monomeric and dimeric *Ssol*-aLhr2 were also shown to have DNA strand annealing activities that are comparable to that of *Tbar*-aLhr2 [16]. Intriguingly, we found that *Tbar*-aLhr2 is less prone to unwind duplexes (Figure 7A) than to anneal nucleic strands (Figure 7B). Indeed, we noted that the in vitro unwinding activity of *Tbar*-aLhr2 is slow. This is also the case for *Msme*-bLhr-HTH which was shown to have a capacity to only slowly dissociate nucleic acid strands (observed after 30 min of incubation) [6]. This can be relevant for the cellular functions of *Tbar*-aLhr2 but it can also mean that our experimental conditions are not optimal. 

ATP binding is proposed to act as a molecular switch from a strand-annealing to an unwinding mode by changing the protein conformation. Here, we showed that *Tbar*-aLhr2 unwinding activity is independent of the presence of ATP. While it might seem surprising, ATP-independent unwinding activities were previously reported for the human SF2 NS3h helicase and the bacterial SF1 RecBCD helicase [45,46]. It was proposed that the energy required for duplex separation is provided by nucleic acid binding and not by ATP binding and hydrolysis. In vivo, it is possible that the balance between unwinding and annealing states is displaced upon interaction with specific protein partners. 

We also investigated the role of Domain 4, a novel structural domain specific to Lhr proteins, and demonstrated that it is essential for *Tbar*-aLhr2 to adopt active conformations. First, we showed that while the ATPase activity is carried by the SF2 catalytic core composed of the RecA1 and RecA2 domains, Domain 4 stimulates ATP hydrolysis (Appendix A). Finally, we found that Domain 4 is essential for *Tbar*-aLhr2 annealing and unwinding activities. The substitution of the highly conserved tryptophan at position 577 in *Tbar*-aLhr2 Domain 4 is sufficient to abolish these reactions (Figure 8). These results are in agreement with those obtained for bacterial *Msme*-bLhr-HTH restricted to the “Lhr core” [9].

Our results underline the high capacity of *Tbar*-aLhr2 to perform nucleic acid strand annealing. This property could be of great importance in hyperthermophile organisms, such as *T. barophilus*, for maintaining nucleic acid duplexes at high temperatures. While there is much that remains to be discovered with respect to the cellular functions of aLhr2 in vivo, we can propose the following roles for Thermococcales aLhr2. First, the detection of *Paby*-aLhr2 in the interaction network of the replication protein A complex (RPA) [19] is consistent with studies proposing that aLhr2 helicases are involved in DNA recombination and repair in *S. solfataricus* and *M. thermautotrophicus* [15,41]. Indeed, RPA that binds ssDNA is crucial for both DNA replication and DNA damage response [47]. This is also coherent with an involvement of Thermococcales aLhr2 in RNA transactions. In *P. abyssi*, RPA was also shown to enhance transcription [19] and to be part of the interaction network of 5′-3′ exoribonuclease aRNase J [20], questioning its involvement in RNA metabolism. The involvement of Thermococcales aLhr2 in RNA metabolism is also supported by our initial observation that *Paby*-aLhr2 was found to be in the protein network of ASH-Ski2 with a high specificity index (Appendix A); ASH-Ski2 is an archaeal specific Ski2-like helicase that forms a complex with aRNase J [20]. Interestingly, RPA is also found in the interaction network of ASH-Ski2 (Appendix A). 

Furthermore, we showed that *Tbar*-aLhr2 is a DNA/RNA helicase able to process DNA:RNA duplexes. The ability of Lhr proteins to process such hybrids was also identified for bacterial *Msme*-bLhr-HTH and *Pput*-bLhr, and for archaeal *Mthe*-aLhr2 helicases [6,7,9,15]. In the cells, DNA:RNA hybrids are often found in the R-loop, a three-stranded structure that harbours a DNA:RNA hybrid and a displaced single-stranded DNA. Controlling R-loop formation and suppression is critical for many cellular processes. While R-loops are often associated with genome instability, DNA damage and transcription elongation defects, mounting evidence suggest that R-loops promote DNA transactions including DNA recombination and repair [48]. Interestingly, RPA was also recently revealed to act as a sensor of R-loops and to regulate RNase H1 in human cells [49]. Moreover, defect in mRNA processing was recently associated with R-loop-dependent genome instability in Eukaryotes [48]. Further physiological and mechanical studies are necessary to determine the function(s) of aLhr2 in Thermococcales cells.

## Figures and Tables

**Figure 1 biomolecules-11-00950-f001:**
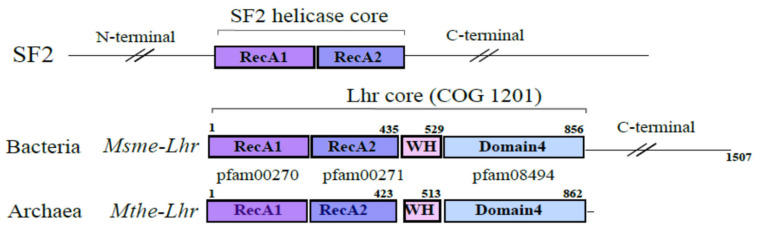
Overall domain organization of SF2 helicase superfamily and Lhr-like subfamily. Lhr proteins of the *Mycobacterium smegmatis* bacterium and the *Methanothermobacter thermautotrophicus* archaeon are shown. The “Lhr-core” (COG1201) is composed of the RecA1 (PF00270) and RecA2 (PF00271) domains of the SF2 helicase core, the Winged-Helix domain (WH) and the Domain4 (PF084494) of unknown function that is specific to the Lhr proteins.

**Figure 2 biomolecules-11-00950-f002:**
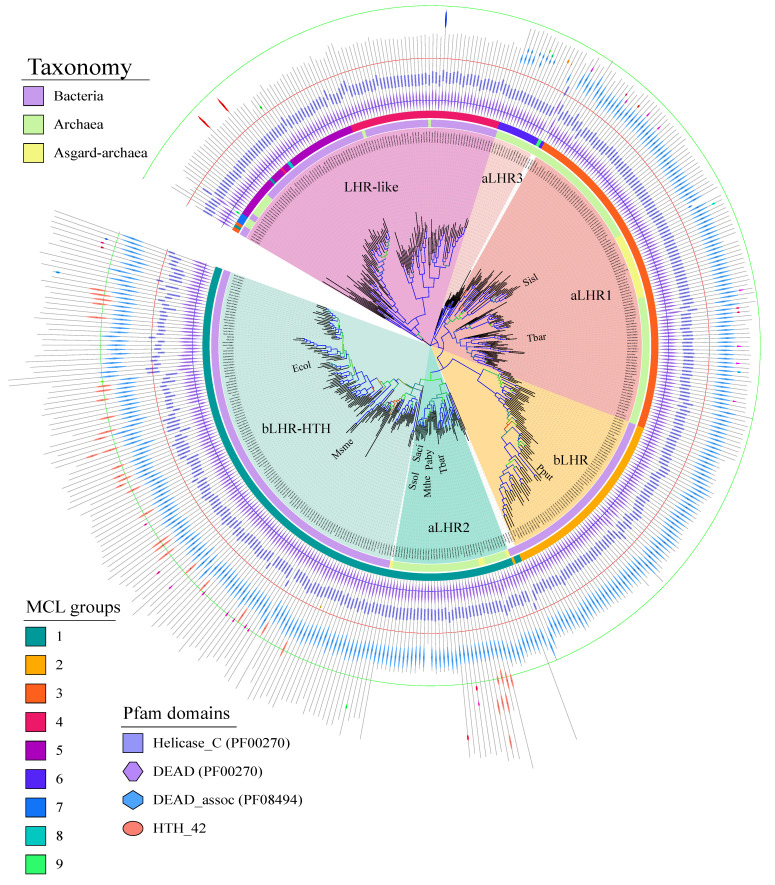
Phylogenetic tree and domain organization of archaeal and bacterial Lhr representative sequences. The tree is rooted to the branch that separates the subtrees Lhr-like and aLhr3. The branches are coloured according to their bootstrap value using a colour gradient from red (bootstrap value of 0) to blue (bootstrap value of 100) with flashy green at midpoint values. The taxonomic origin of the sequences is shown in the first outer ring: purple for Bacteria, light green for all Archaea except for Asgard, shown in yellow. The MCL subclasses obtained with an IF of 4 are shown on the second outer ring and colour-coded as indicated in the legend panel “MCL groups”. To highlight the different subfamilies, their corresponding subtrees are coloured and the sequences to which the location on the tree and the MCL classification do not correspond are left white. The locations of reference sequences are indicated at their respective leafs (*Tbar*, *Sisl*) or at the leaf of their representative medoid (*Msme*, *Ecol*, *Pput*, *Ssol*, *Saci*, *Mthe*, *Paby*). Pfam motif architecture for each member is specified at the circumference of the tree. Their colour code is given in the legend panel “Pfam domains”. The sequence identifiers for each subtree are in Appendix A. The tree display was obtained with online iTOL [29] (https://itol.embl.de/; accessed on 3 May 2019).

**Figure 3 biomolecules-11-00950-f003:**
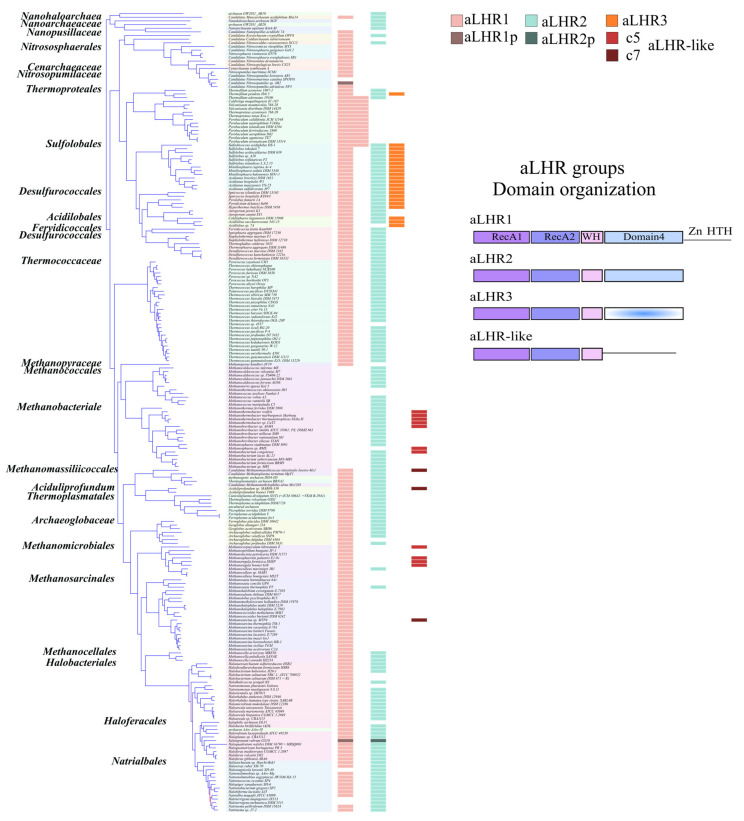
Distribution and domain organization of Lhr groups in Archaea. Left panel: the species tree of the archaeal genomes was deduced from sets of 120 concatenated sequence alignments with fasttree [30] using LG + GAMMA models and support values determined using 100 non-parametric bootstrap replications (lower bootstrap supports are indicated by red branches). The tree was rooted on DPANN Archaea according to [31]. NCBI taxonomy was reported at the order or family level. The distribution of aLhr groups is displayed as bar charts, whose width is proportional to the number of paralogues in each genome (0, 1 or 2). The darker marks in the aLhr1 and aLhr2 column indicate the presence of pseudogenes and are figured as aLhr1p and aLhr2p. The c5 and c7 correspond to archaeal aLhr-like proteins. Right panel: the domain architecture is represented for each archaeal aLhr family. The SF2 helicase core is shown in light and dark violet, WH domain in pink and Domain 4 in light blue. aLhr1 architecture shows an additional C-terminal domain containing a conserved Zn-finger like and HTH motifs; aLhr2 is restricted to the “Lhr core”; aLhr3 has a deteriorated Domain 4 (in gradient blue) and aLhr-like misses a Domain 4.

**Figure 4 biomolecules-11-00950-f004:**
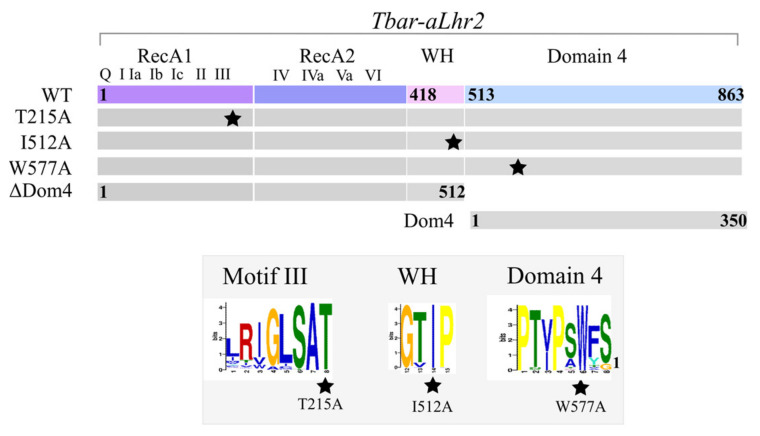
Domain organisation of wild-type *Tbar*-aLhr2 and derivatives. The colour code is as in Figure 3. Punctual mutations are represented by a star. The weblogos at the site of mutation are shown below.

**Figure 5 biomolecules-11-00950-f005:**
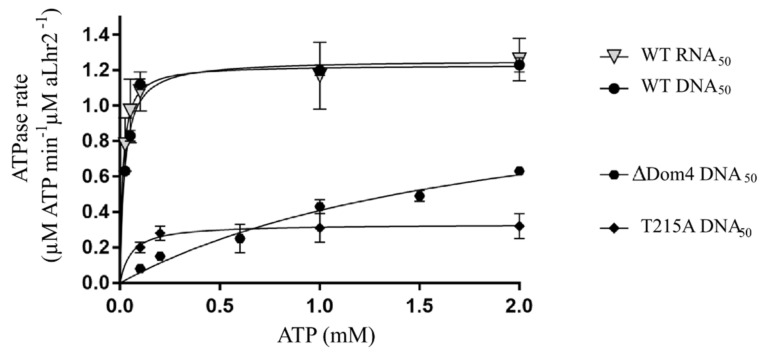
ATPase activity of wild-type *Tbar*-aLhr2 and of ΔDom4 and T215A derivatives in the presence of nucleic acid molecules. The apparent Michaelis dissociation constant (Km) of *Tbar*-aLhr2-WT for the DNA_50_ and RNA_50_ molecules are 23 ± 1 µM and 13 ± 1 µM, respectively. See also Appendix A.

**Figure 6 biomolecules-11-00950-f006:**
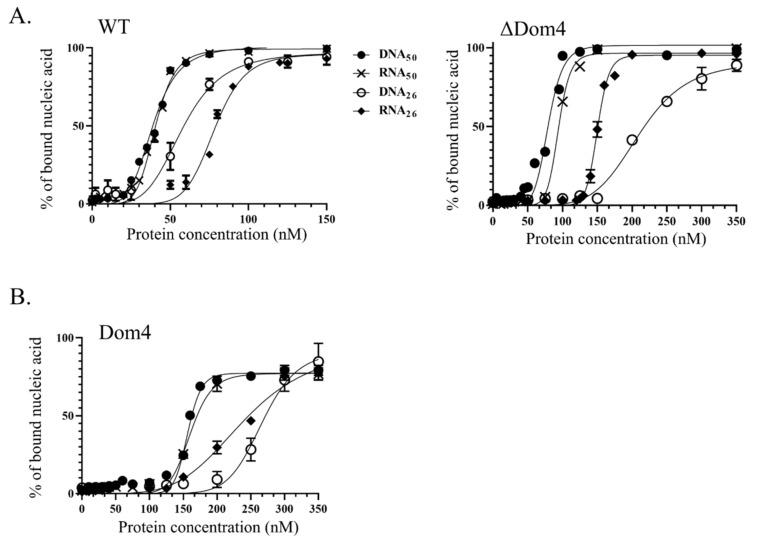
Binding affinities of wild type (WT) *Tbar*-aLhr2 and derivatives for single-stranded nucleic acids. (**A**) Using WT and ΔDom4 *Tbar*-aLhr2, the percentage of nucleoprotein complex formed after 15 min of incubation was plotted against the protein concentrations. The experiments were carried with RNA_50_, RNA_26_, DNA_50_ and DNA_26_ substrates (Appendix A). Three independent experiments were performed in each condition. (**B**) Identical assays were performed with a recombinant protein corresponding to Domain 4 alone.

**Figure 7 biomolecules-11-00950-f007:**
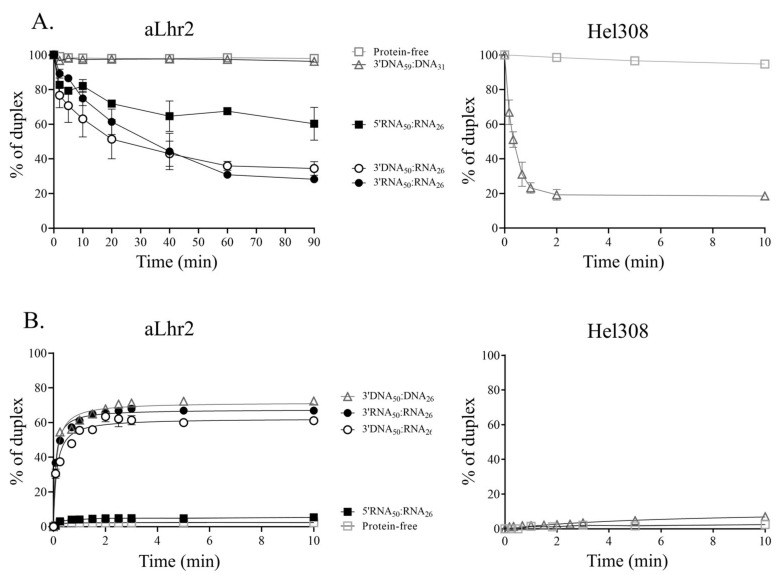
Unwinding and annealing activities of wild type *Tbar*-aLhr2. (**A**) Left panel: Kinetics of strand dissociation by *Tbar*-aLhr2 in the presence of ATP are shown for 3′ overhang duplexes (3′DNA_59_:DNA_31_, 3′DNA_50_:RNA_26_ and 3′RNA_50_:RNA_26_) and 5′ overhang duplex (5′RNA_50_:RNA_26_); right panel: Kinetics of strand dissociation by *Paby*-Hel308 are shown for 3′DNA_59_:DNA_31_ duplexes; (**B**) Kinetics of strand association in absence of ATP by *Tbar*-aLhr2 are shown for 3′ overhang duplexes (3′DNA_50_:DNA_26_, 3′DNA_50_:RNA_26_ and 3′RNA_50_:RNA_26_) and 5′ overhang duplex (5′RNA_50_:RNA_26_); Three independent experiments were performed in each condition.

**Figure 8 biomolecules-11-00950-f008:**
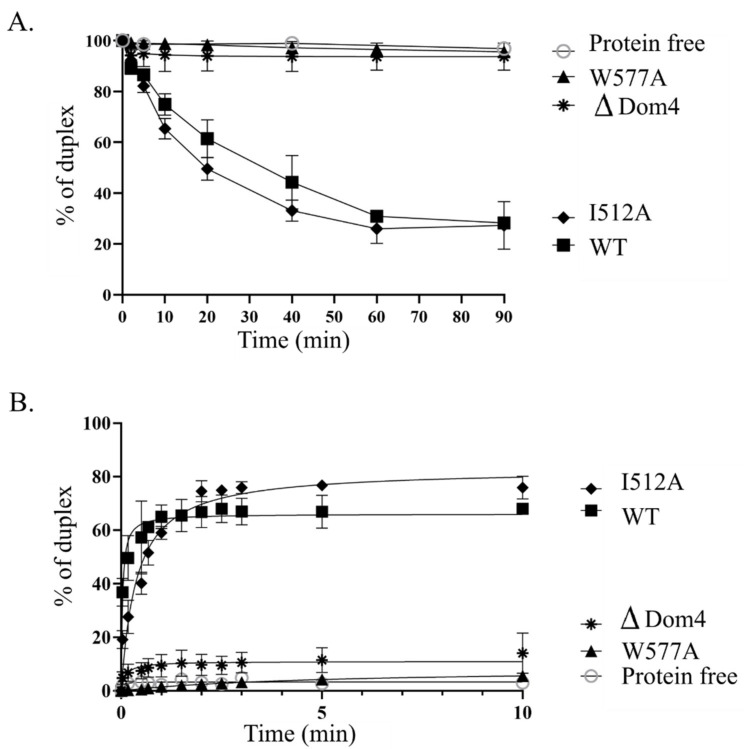
Domain 4 is critical for *Tbar*-aLhr2 unwinding and strand-annealing activities. (**A**) Kinetics of unwinding reaction (performed as in Figure 7A) of *Tbar*-aLhr2-ΔDom4, *Tbar*-aLhr2-W577A and *Tbar*-aLhr2-I512A using the 3′RNA_50_:RNA_26_ substrate are shown; (**B**) Kinetics of strand-annealing reactions (performed as in Figure 7B) of *Tbar*-aLhr2-ΔDom4, *Tbar*-aLhr2-W577A and *Tbar*-aLhr2-I512A to form 3′RNA_50_:RNA_26_ duplexes are shown.

**Table 1 biomolecules-11-00950-t001:** Apparent dissociation constant (Kd in nM) derived from data shown in Figure 6 (n.d. means not determined).

Substrates	WT	ΔDom4	Dom4	Hel308
DNA_50_	39 ± 1	54 ± 1	162 ± 5	32 ± 2
DNA_26_	58 ± 2	211 ± 9	277 ± 16	n.d.
RNA_50_	41 ± 1	69 ± 1	161 ± 5	38 ± 1
RNA_26_	80 ± 2	150 ± 1	241 ± 17	n.d.
3′DNA_59_:DNA_31_	41 ± 2	n.d.	n.d.	23 ± 1
3′RNA_50_:RNA_26_	39 ± 2	n.d.	n.d.	n.d.

## Data Availability

The datasets generated and/or analysed in this study are included in this published article and its Appendix A. They are also available from the corresponding authors upon reasonable request.

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
