# Peer review of "Phylogenetic Diversity of Lhr Proteins and Biochemical Activities of the Thermococcales aLhr2 DNA/RNA Helicase"

_biomolecules, 2021, doi:10.3390/biom11070950_

Round 1
Reviewer 1 Report
The manuscript by Hajj et al. describes the purification and characterization of aLhr2 helicases. In general, I have little enthusiasm for this manuscript being published here since the data and subsequent conclusions represent an incremental advancement in a deeper understanding of the cellular function and mechanism of DNA and RNA dependent helicases. In this regard, many of the conclusions drawn by the authors are based upon very qualitative data. Provided below are specific examples that must be addressed with more experimental details.
- In Figure 5, the authors claim that aLhr2 has a preference for DNA/DNA substrates as judged by simply comparison of the time courses in ATP hydrolysis. In particular, the authors focus on the plateau in ATP hydrolysis as a function of nucleic acid substrates. This is an extremely qualitative assessment. A more appropriate approach is to measure kcat and Km values for ATP with each nucleic acid substrate. These data would allow the reader to more accurately gauge if there is truly a preference for nucleic acid substrate, and if so, whether this preference is caused by an increase in kcat, a decrease in Km, or both. This analysis would significantly aid in other sections of the manuscript in which the authors assess nucleic acid unwinding activity with different nucleic acid substrates.
- The nitrocellulose binding data used to measure the dissociation constants (Kd) for wild type and mutant aLhr2 are suspect. In all of the displayed titrations, there are, at most, only two (2) points that lie within the range where an accurate Kd value can be measured. In other words, there are not enough data points in the range of 10% to 90% binding to accurately define Kd values. As such, the reported errors are far too low from the primary titration plots.
- There are significant problems with Table 2. First, the title is inaccurate. Secondly, the authors do not provide units for the measured Kd values. Finally, more work is needed to investigate the ability of Dom4 to bind nucleic acid. Based upon their data, the authors propose that Dom4 possesses a low affinity binding site for nucleic acid. However, this conclusion is not supported by the data using long nucleic acids (DNA50 and RNA50) as there is a minimal effect (<2-fold) on the measured Kd values. Clearly more work is needed to verify this conclusion.
Author Response
"Please see the attachment."

Reviewer 2 Report
This manuscript comprises two parts: Bioinformatics analysis and biochemical characterization. In the bioinformatics part, authors performed in depth-phylogenomics analyses of Lhr helicases in Archaea. In the biochemical part, authors characterized Tbar-aLhr helicase, including its ATPase, binding, unwinding, and strand annealing. It is an interesting study for Lhr helicases in archaea and bacteria. Here are some issues should be addressed before publication.
Here are my comments:
- Title: “Towards the Function…”, when we talk about function, usually means biological function, at least cellular function, while no such function of aLhr2 was done in current report; therefore, I feel the title, particularly the word function, is overstated.
- Line 55, Fig. 1, might include the domain structure of aLhr1 and aLhr2.
- Line 202, pET15b was used in this study?
- Page 14 results 3.2.1 Fig. 5 ATPase: why ssDNA, not dsDNA, was used?
- Page 15 results 3.2.2 Fig. 6 nucleic acid binding assays, again, why not dsDNA? Since Tbar-aLhr2 prefers dsRNA and Hel308 prefers dsDNA for unwinding, have you checked their binding ability?
- Line 525, Fig. 7 not Fig 1. Panel C was not described. Panel A, please indicate what’s substrate used here. A-B, a solid negative control should be one of T215A or I512A, NOT protein-free. Given reaction was performed at 65ËšC for 90 min, a solid negative should be included.
- Page 20, Fig. 8. Panel A: please label and describe what’s kind substrate used here. Notably, there is annealed duplex in protein-free reaction.
- Line 562, “in presence of ATP, AMP-PNP and ATP-gama-S, Tbar-aLhr2 lost its annealing capacity.” It is reasonable that ATP will fuel unwinding reaction, any thoughts why non-hydrolysable ATP analogues will inhibit/abolish annealing activity?
Note: this reviewer is not an expertise in phylogenomic studies; thus, I will leave this work to other reviewer(s). Nevertheless, here are some questions, may be non-scientific:
- Since “Lhr-Core” contains two RecA domain, a WH motif and a Domain 4, For example, Line 157, “i.e., the RecA1 and RecA2 regions (Figure1)”. Line 306: “alignments were performed with the SF2 helicase core domain”, why not “alignments were performed with the LHR core domain”?
- How authors gathered Lhr helicases from archaea and bacteria genomes should be described straightforward and clearly. Fig. 2 and 3, obviously it is hard to see the small font, hopefully it is readable/informatic enough.
- Since this work focuses on Lhr helicases in Archaea, why bacteria was included in Fig. 2.
Author Response
"Please see the attachment.

Reviewer 3 Report
Towards the function of aLhr2 helicases in Archaea
Hajj et al. 2021
Nucleic acid helicases, particularly superfamily 2 (SF2) helicases, are ubiquitous throughout life and serve important and sometimes indispensable functions pertaining to DNA repair, DNA replication, chromatin remodeling, RNA maturation, ribosome synthesis, viral defense, and more. This manuscript provides a perspective of the SF2 Large helicase related family of helicases well-conserved (but poorly characterized) across Bacteria and Archaea. While conservation implies biological significance, it does not inform function, and thus the authors provide taxonomically information on a large suite of Lhr2 proteins and briefly biochemically analyze an Lhr helicase (aLhr2) from the euryarchaeon Thermococcus barophilus. The findings presented unfortunately do not rise beyond observational notes, and thus the impact of the findings on the community is deemed rather low and likely insufficient to warrant publication at this time. This limited scope abrogates our holistic understanding of Lhr helicases and diminishes the impact of the manuscript on the SF2 field. The manuscript can be best summarized on line 684: “Our results alone do not allow inferring of cellular roles for Thermococcales aLhr2”.
Significant concerns
1 - The manuscript references previous studies wherein Lhr2 homologues were identified within RNA-based interactomes. The manuscript is thus written to guide the readers towards a RNA-focused or RNA-relevant role for Lhr2 proteins in vivo. The interactome studies and biochemical assays are derived from unique species, and given the dispersity of Thermococcale Lhr2 proteins in the taxonomic analyses, little evidence is provided that supports a role for Lhr2 in RNA transactions. Many helicase and translocases have in vitro activities on both RNA and DNA, but this does not define in vivo function.
2 - While the taxonomical distributions are certainly in-depth, it is hard to determine what is important from the data introduced, and the figure presentation causes more confusion than clarity. The resolution and size of the figures impairs any meaning to be derived from the relationships proposed. The figures must be altered. Figures in general are they very low resolution. This makes it hard to absorb the vast amount of data presented in the taxonomical studies. Figure 7 is incorrectly labeled Figure 1. As written, the division of clades does not provide evidence of biological function. If there are 3 clades of archaeal Lhr2 proteins, should one assume three functions? or overlapping functions? Similarly disappointing information is provided for the proposed clades of bacterial Lhr2 proteins.
3 - The helicase and annealing assays appear to provide some insights into the function of the enzyme, but the slow rates of these activities call into question their biological significance- something which is scarcely addressed in the manuscript. The authors limit biochemical analyses to one isoform of the aLhr2 helicases, when various are found throughout Domains, and even within the euryarchaeal clade. This limited scope abrogates our holistic understanding of Lhr helicases and diminishes the impact of the manuscript on the SF2 field. Any proposed role for the enzyme is incredibly speculatory due to the limitations of the data. aLhr2 from T.barophilus, and the other isoforms found in euryarchaea at least, should be further explored experimentally before the impact of this manuscript warrants publication.
4 - In Materials & Methods, 2.1 Building Lhr-type dataset, the authors go through an extensive protocol to filter archaeal Lhr-like proteins into classes/clades. The end result was that eight unique sequences were discarded and in line 143, the authors state that the two Lhr sequences from T. barophilus they are interested in were not retained in their reduction protocol and they added them back in. No explanation on how the authors were able to justify adding the two Lhr sequences back into their dataset was provided and thus the biochemical results presented cannot be logically assigned to any class of proteins? Did any of the other sequences removed have any constraint similarities to the two T. barophilus sequences, and if so, how do the authors justify adding these back in and not the others?
5 – The discussion text is more concisely written than the results section yet the same information is presented. The first half of the discussion appears to mirror closely the early-main body of the manuscript. It does, however, appear to describe the taxonomical distribution of aLhr2 enzymes a lot better than in the main body of the manuscript. The same information does not need to be repeated back to back to artificially extend the length of the text.
Minor points
- Line 37-40: The authors appear to gloss over the SF2 field, which is likely the field that would benefit the most from this manuscript. It may be important to note that many SF2 helicases do not separate strands of nucleic acids, and many do not contain translocase activity.
- Figure 2: Very low resolution and very confusing. Consider a different way of displaying this. Maybe a simple table. The figure legend is entirely uninformative.
- Figure 3: See Figure 2 notes.
- Line 430-436: In Thermococcales, these enzymes appear quite diverse. The authors are studying the barophilus form of the enzyme because the P.abysii (the organism from which the interactome network was formed for the basis of this paper) enzyme is toxic. Does this interactome hold up across species?
- Table 1: The effect of the specific sequence of nucleic acids on these assays is barely mentioned in the text and is not varied to determine sequence-specific effect. It breaks up the text too much and adds little to the manuscript, and may be best served as a supplementary figure.
- Figure “1” (7): These helicase activities are on the order of hours with 50 fold excess enzyme whereas other SF2 helicases can unwind substrate in just 3 to 10-fold excess amounts in just minutes. How relevant is this helicase activity?
- Figures 8 and 9: With two nucleic acid binding sites, how is the determination made between the enzyme simply localizing the separate strands of nucleic acid allowing them to anneal faster, versus using SF2-like functions to specifically anneal strands? It looks like the equilibrium of annealing:separating favors annealed strands in these conditions, but when ATP is involved nothing happens (lines 562-565)- this seems biologically irrelevant.
Author Response
"Please see the attachment."

Round 2
Reviewer 1 Report
This is a much improved revised version. However, there are two (2) experimental issues that remain to be addressed prior to acceptance. The first issue is with respect to the Michaelis-Menten plots provided in Figure 4. The data for the WT helicase with DNA and RNA as well as that for the T215A mutant is not accurate since the concentrations of ATP used to measure rates in ATP hydrolysis are at or above the report Km value. The authors must perform additional experiments to examine rates in ATP hydrolysis at lower ATP concentrations in order to accurately define these Km values in addition to determining if there is any cooperativity in ATP binding and/or hydrolysis. This is especially important in light of the data provided in Figure 6 which is the basis for my other concern. In this case, the authors claim, based on size exclusion chromatography, that the helicase is a monomer. While this is likely true, the authors have not assessed whether or not the helicase remains as a monomer in the presence of nucleic acid or if it assembles as an oligomer. This is essential information as the authors show cooperativity in helicase binding to nucleic acid in Figure 6. They conclude that this represents multiple monomers binding to nucleic acid. However, it is also possible that this cooperativity represents oligomerization of the helicase that is dependent on nucleic acid.
Reviewer 2 Report
no further questions.
Author Response
"Please see the attachment.

Reviewer 3 Report
We thank the authors for reading and considering critiques of the submitted manuscript regarding Lhr helicases and submission of a revised version, now titled Phylogenetic diversity of Lhr proteins and biochemical activities of the Thermococcales aLhr2 DNA/RNA helicase. The updated title more accurately reflects the research conducted and limits the scope to Thermococcales. The authors have made additional, albeit minor, changes in response to the prior reviews. The introduction (lines 31-46) and Figure 1 more clearly relate the manuscript to the superfamily 2 helicase field which is most likely to benefit from the work. The more pointed explanation of the procedures permitting enzymes to be ‘adding back’ to the taxonomical analysis which were the subject of this study improves the study. The helicase activity assays now have more clarity after reporting a shorter time course and more visibly comparing the activity to that of Hel308- although a clearer schematic of the ‘strandedness’ and ‘polarity’ of each substrate used would still greatly improve the figure now that the oligonucleotide sequences have been moved to supplemental material. The discussion is improved, offering a more generalized overview and pointing to less conclusions from the limited data. The manuscript could be described as more cautious in its finding.
While these revisions improve the manuscript, the authors did not address suggestions from multiple reviewers to add additional experiments before the advancements of understanding presented are significant enough to warrant publication. The manuscript lacks impact given that essentially no new experimental evidence is presented to define the biological roles of the enzymes under study. With the experimental procedures and assays already established, it is reasonable to attempt to purify and examine minimally one other enzyme from another Lhr group (i.e. aLhr1, aLhr2, aLhr3), as well as a form from a mesophilic archaeon to quickly test they hypothesis of strand annealing activities being relevant in a thermophilic organism. Without investigation of another form of the archaeal Lhr enzymes, the current understanding and contribution to the field is flawed in that assumptions will be made about all Lhr helicases based on data from one hyperthermophilic form of the enzyme. Failure to define a biological importance of these enzymes through some combination of microbiology, genetics, biochemistry and/or enzymology is a fatal flaw of this work.
Until a more representative population of Lhr helicases are examined biochemically the research does little to add to our understanding of Lhr helicases in general. Rather, the manuscript only incrementally advances understanding of one specific Lhr-helicase from one specific organism which happens to encode two groups of Lhr proteins and exists at temperatures vastly different to the bulk of Lhr enzymes characterized taxonomically. Until these conflicts are resolved, the manuscript does not warrant publication.
Minor issues:
Line 94/Table S1: While the authors refute suggestion an RNA-relevant in vivo role for the Lhr helicases in the manuscript, multiple references are made to an interactome study where the Lhr helicases were found in a network of RNA-metabolism enzymes. The authors fail to mention that there are just about as many DNA replication and DNA metabolism enzymes in this network.
Author Response
"Please see the attachment.
